

# Deep reinforcement learning task scheduling method based on server real-time performance

Jinming Wang, Shaobo Li, Xingxing Zhang, Fengbin Wu and Cankun Xie

State Key Laboratory of Public Big Data, Guizhou University, Guiyang, Guizhou, China

## ABSTRACT

Server load levels affect the performance of cloud task execution, which is rooted in the impact of server performance on cloud task execution. Traditional cloud task scheduling methods usually only consider server load without fully considering the server's real-time load-performance mapping relationship, resulting in the inability to evaluate the server's real-time processing capability accurately. This deficiency directly affects the efficiency, performance, and user experience of cloud task scheduling. Firstly, we construct a performance platform model to monitor server real-time load and performance status information in response to the above problems. In addition, we propose a new deep reinforcement learning task scheduling method based on server real-time performance (SRP-DRL). This method introduces a real-time performance-aware strategy and adds status information about the real-time impact of task load on server performance on top of considering server load. It enhances the perception capability of the deep reinforcement learning (DRL) model in cloud scheduling environments and improves the server's load-balancing ability under latency constraints. Experimental results indicate that the SRP-DRL method has better overall performance regarding task average response time, success rate, and server average load variance compared to Random, Round-Robin, Earliest Idle Time First (EITF), and Best Fit (BEST-FIT) task scheduling methods. In particular, the SRP-DRL is highly effective in reducing server average load variance when numerous tasks arrive within a unit of time, ultimately optimizing the performance of the cloud system.

# INTRODUCTION

Cloud computing is a technology and service model that provides computing resources such as computing power, storage, and networks over the Internet. It permits users to acquire and leverage computing resources through the cloud service providers' infrastructure without worrying about the management of the underlying hardware and software. Among them, virtualization technology is critical in cloud computing (*Tong et al., 2021*). Cloud computing can achieve higher efficiency, scalability, and flexibility through virtualization technology. Hence, the applications of cloud computing

Corresponding author
Shaobo Li, lishaobo@gzu.edu.cn

are pervasive. However, many problems still need to be settled in cloud computing. The task scheduling problem is one of vital in cloud computing. This problem involves the efficient allocation and management of limited computing resources to perform various types of tasks. The task scheduling approach is currently one of the effective methods to address resource constraints in complex cloud environments, and it is critical to promote the quality of Quality of Service (QoS) indicators like resource utilization and service delay.

It is tough for traditional scheduling algorithms to show excellent performance in cloud resource allocation and task scheduling due to the complexity of cloud environments, the dynamics and diversity of resource demands, and the heterogeneity of resources. This limitation primarily arises from their inherent constraints, making them unable to meet the dynamic requirements of cloud resource environments, thereby impacting user experience and performance. In contrast, reinforcement learning can flexibly adapt to changing workloads and resource status by learning and optimizing scheduling decisions. The learning framework with the reward mechanism enables the system to make optimal scheduling decisions under real-time requirements, allowing it to be adjusted to the ever-changing cloud computing environment. Therefore, applying reinforcement learning to cloud task scheduling is highly appropriate.

When scheduling tasks in a cloud environment, since the diversity and dynamics of task demands, along with the continuity of tasks, the server will occupy some resources after accepting and processing task requests. With the arrival of subsequent tasks, previously arrived tasks impose a particular load on the server, influencing service time (*Delasay et al., 2019*). An increase in server load usually leads to delays in response time, that is, an increment in the processing time of client requests, affecting the user experience. Simultaneously, the performance of web servers is immediately influenced by the load directed towards them (*Jader, Zeebaree & Zebari, 2019*). Server performance directly affects the efficiency of cloud task processing, resource utilization, and system stability. Consequently, in the decision-making process of implementing cloud task scheduling using reinforcement learning methods, it is necessary to comprehensively consider the interaction between load and performance. This can accurately evaluate the real-time processing capabilities of the server, thereby rationally allocating tasks to servers with better performance. In this way, tasks can be processed faster, task response times can be reduced, and cloud resource utilization and overall system performance can be improved.

We propose a deep reinforcement learning task scheduling method based on server real-time performance (SRP-DRL). It introduces a real-time performance-aware strategy to enhance the cloud scheduling environment awareness of the deep reinforcement learning (DRL) model. The mathematical model of the reinforcement learning architecture can conveniently represent the scheduling process dynamics, and as a learning-based algorithm, the DRL algorithm has high model generalization capability (*Chen et al., 2023b*). SRP-DRL aims to optimize the load balancing capability of servers and to consider the effect of server load on server performance on top of server load to perfect the state information fed back from the environment for improving the decision-making capability of the model. The overall contributions of our research are as follows.

1) Utilizing power function models with nonlinear relationships and load thresholds to establish a real-time performance platform model to accurately model the impact of load on performance, thereby enhancing the precision and efficiency of cloud task scheduling.

2) By introducing a real-time performance-aware strategy, the influence of task-generated load on server performance is considered, enhancing the DRL model's ability to perceive environmental states.

3) Propose a DRL task scheduling method built on server real-time performance to reduce task response time under latency constraints and enhance the load balancing capability of servers.

4) SRP-DRL has undergone extensive evaluation, comparing its performance with several commonly used baseline task scheduling schemes on various dimensions and metrics, including Random, Round-Robin, Earliest Idle Time First (EITF) and Best Fit (BEST-FIT) task scheduling methods. Experimental results demonstrate the superiority of SRP-DRL over the mentioned baseline task scheduling methods.

The following structure characterizes the remainder of the article. "Related Work" delves into related work on task scheduling methods for optimizing load balancing. "System Model" introduces the system model and depicts the relevant system assumptions. In "The Proposed SRP-DRL", we illustrate the design details of the DRL task scheduling method (SRP-DRL) based on server real-time performance. "Experiment" presents and analyzes the experimental outcomes across various dimensions and indicators, and confirms that the proposed SRP-DRL is superior to other comparative baseline task scheduling methods. Finally, we conclude this article and propose improvements for future work in "Conclusion".

## RELATED WORK

The real-time performance of the server will affect the allocation decisions of cloud tasks, thereby affecting the effect of load balancing. Although traditional load balancing algorithms (*Jafarnejad Ghomi, Masoud Rahmani & Nasih Qader, 2017*; *Shafiq, Jhanjhi & Abdullah, 2022*) such as Round Robin, Min-Min, Opportunistic Load Balancing, and Max-Min are simple to implement and perform well in relatively stable systems. Yet these algorithms are limited when considering server real-time loads, performance variations, and adaptability. To solve the adaptability problem of traditional load-balancing algorithms, many scholars have studied swarm intelligence optimization algorithms. Commonly used swarm intelligence optimization load-balancing algorithms include Artificial Bee Colony (*Kruekaew & Kimpan, 2022*), Ant Colony Optimization (*Shi, Hu & Lu, 2021*), Particle Swarm Optimization (*Dubey & Sharma, 2021*), and Genetic Algorithms (*Zhou et al., 2020*), *etc*. These algorithms are highly adaptable and suitable for complex environments, but they usually have problems such as poor adaptive capabilities, non-real-time decision-making, and insufficient flexibility. Thus, to solve the problems existing in the above-mentioned intelligent optimization algorithms, reinforcement learning is introduced to achieve load balancing of cloud task scheduling. Owing to the complexity of

the cloud environment, the dynamics of task requirements, and the characteristics of DRL suitable for solving complex optimization problems in high-dimensional state spaces (*Liu et al., 2022*), DRL has demonstrated outstanding performance in processing cloud tasks. Indeed, optimizing cloud task scheduling for load balancing using DRL has attracted significant attention and research. The following focuses primarily on the research work related to DRL optimization for load balancing and is illustrated by the two metrics: makespan and response time.

**Makespan**. *Cheng, Li & Nazarian (2018)* proposed an approach built on DRL, which integrates resource provision and task scheduling systems. *Tong et al. (2020)* suggested a deep Q-learning task scheduling algorithm. It integrates the strengths of Q-learning algorithms and deep neural networks to decrease the task completion time. *Dong et al. (2020)* presented a task scheduling algorithm depending upon DRL, which dynamically allocates priority tasks on cloud servers and optimizes task execution time. *Swarup, Shakshuki & Yasar (2021)* developed a task scheduling algorithm according to clipped double-deep Q-learning to diminish task execution time and costs. *Hu, Tu & Li (2019)* introduced a task scheduling framework, that employs Monte Carlo Tree Search (MCTS) for task scheduling and trains a DRL model to instruct the scaling and launch of MCTS. When considering tasks reduce the completion time of complicated jobs under dependencies and heterogeneous resource requirements. *Wu et al. (2018)* proposed an adaptable directed acyclic graphs task scheduling algorithm DRL-based to reduce task completion time. *Mangalampalli et al. (2023)* introduced a task scheduling method based on deep Q-learning networks (DQN) to optimize completion time and the percentage of service level agreement (SLA) violations. *Grinsztajn et al. (2021)* developed a dynamic task scheduling algorithm for directed acyclic graphs, which merges graph convolutional networks with an advantage actor-critic algorithm to shorten task completion time. *Dong et al. (2023)* introduced an adaptive fault-tolerant workflow scheduling framework that combines double deep Q networks (DDQN) to reduce task completion time while achieving fault tolerance. *Mangalampalli et al. (2024)* provided a multi-objective workflow scheduling algorithm based on DRL, which captures data feedback such as dependencies and task priorities to reduce completion time. *Chen et al. (2023a)* suggested a cloud computing heterogeneous workflow collaborative scheduling method combined with DRL to optimize workflow completion time under task execution continuity constraints. *Cao et al. (2024)* proposed a task scheduling method depending on the graph attention network and DRL to minimize the completion time of user tasks. Besides, a multi-action and environment-adaptive proximal policy optimization algorithm is presented by *Li et al. (2024)*. In this work, a joint task scheduling and resource allocation method was designed to reduce request completion time.

**Task response time.** *Jyoti & Shrimali (2020)* proposed a dynamic resource allocation approach that depends upon load balancing and service proxy to solve the problem of performance reduction of traditional methods in dynamic resource allocation. The local user agent in the method uses multi-agent DRL-dynamic resource allocation to predict user task requests to reduce the response time. *Baek et al. (2019)* presented an offloading method rooted in reinforcement learning to solve the problem of different computing

capabilities of fog nodes, which can keep the convergence of the algorithm in polynomial time and improve the load balancing problem under the restriction of achieving minimum delay in the fog network. *Gazori, Rahbari & Nickray (2020)* introduced a dual deep Q-learning scheduling algorithm using target networks and experiences replay strategy, which is beneficial to reducing service latency. *Rjoub et al. (2021)* proposed four approaches for automatic task scheduling—reinforcement learning, DQN, recurrent neural network long short-term memory (LSTM), and DRL integrated with LSTM. Experiments on a real dataset from the Google Cloud Platform show that these methods can automate workload scheduling and reduce task waiting time. *Chen et al. (2019)* provided the collaborative mobile edge computing intelligent resource allocation framework, which is built on the multi-task DRL algorithm trained of self-play training to shorten service delays. *Ran, Shi & Shang (2019)* employed a deep deterministic policy gradient network to find the optimal distribution solution of tasks that meets the SLA requirements, reduce the average response time of tasks and improve load balancing between virtual machines. *Cheng et al. (2023)* introduced proposed an improved task scheduling strategy optimization algorithm for the asynchronous advantage actor-critic (A3C). They used a convolutional neural network to improve the network structure of A3C and adopted an asynchronous multi-thread training method to decrease task response time and system energy consumption in the edge-cloud collaborative environment. *Sun, Yang & Lei (2022)* provided a task scheduling algorithm according to optimized DRL in heterogeneous computing environments. They used double Q-learning to enhance the primal DQN and reduce average task response time and the standard deviation of machine CPU utilization. *Pang et al. (2024)* developed a vehicle application task offloading framework, using DAG to model task dependencies, DDQN, and specific mobility management strategies to reduce task delays. *Huang et al. (2023)* presented a DRL-based computing offloading and resource allocation algorithm, which dynamically learns the optimal computing offloading and resource allocation scheme by adapting to the network to reduce task processing delays. *Wang et al. (2024)* introduced an IoT application scheduling algorithm integrated into DRL to optimize the response time of heterogeneous IoT applications. *Farimani et al. (2024)* reported a task offloading algorithm depending on Rainbow to reduce the average task delay by effectively integrating the computing resources of edge servers. Moreover, a preemptive cloud job scheduling method based on DRL is proposed by *Cheng et al. (2024)*, which improves the training of scheduling strategies through an effective preemption mechanism to meet the user's expected response time.

Although these studies mentioned above have utilized DRL to optimize relevant indicators of load balancing, they have yet to consider the impact of server load generated by task execution on server performance and thus on task response time. Hence, when using DRL to realize cloud task scheduling load balancing, the interplay of load and performance must be considered to enhance the perceptual capability of the DRL model, thereby improving the performance of DRL in realizing cloud task scheduling load balancing.

# SYSTEM MODEL

In this section, we use formal and mathematical forms to model the problem and give the relevant problem assumptions and definitions. In Table 1, we present the pivotal notations utilized in this section for clarity in subsequent discussions.

## Task analysis model

The user sends a task request to the cloud platform for handling, and the task information corresponding to user requests is stored in a task queue to form a user task request queue. A task request queue contains a series of tasks, which can be expressed as $Q = \{T_1, T_2, \cdots, T_n\}$, where $T$ represents a task, and $n$ means on number of tasks in the task request queue. Tasks do not interfere with each other, cannot be preempted, and can be executed in parallel when meeting the current virtual machine load and own resource requests. An essential attribute of task information is the task length. The task length L is subjected to a normal distribution with mean $\mu$ and standard deviation $\sigma$, that is, $L \sim N(\mu, \sigma^2)$. Of course, the arrival time of tasks $T^a$, is also a vital attribute of task information, which determines the order in which the DRL model allocates virtual machines to tasks. As a classic distribution, the Poisson distribution is widely used in queueing theory and can be used to describe the pattern of task arrivals. Hence, the arrival time $T^a$ of tasks defers to the Poisson distribution with parameter $\lambda$, that is, $T^a \sim P(\lambda)$. At the same time, the response time $T^{res}$ required for task execution on the cloud platform needs to be no greater than the deadline $T^D$ to ensure that the task can be responded to and processed normally. Tasks with a response time exceeding $T^D$ will be considered failed tasks during runtime performance parameter evaluation.

$$T^{res} = T^w + T^e \leq T^D \tag{1}$$

where $T^w$ represents the waiting time required to fulfill task execution, and $T^e$ is the execution time of the task. The waiting time $T^w$ and the execution time $T^e$ will be influenced by the scheduling decisions made by the DRL model.

To validate the robustness of the proposed task scheduling method and to show the diversity of task requests, we randomly sample the resource requests of tasks for CPU, memory and disk I/O in the range of $[0, \theta)$. As a result, in this scheduling model, each task is represented as

$$T_i = \left\{l_i, T_i^a, reqC_i, reqM_i, reqIO_i, T_i^D\right\} \tag{2}$$

$$subject\ to \quad reqC_i, reqM_i, reqIO_i \in [0, \theta) \tag{2a}$$

where $l_i$ is the task length of the $i$-th task, $T_i^a$ is the arrival time of the $i$-th task, $reqC_i$ is the CPU resource request of the $i$-th task, $reqM_i$ is the memory resource request of the $i$-th task, $reqIO_i$ is the disk I/O resource request of the $i$-th task, and $T_i^D$ is the deadline of the $i$-th task, $\theta$ denotes the load threshold of the virtual machine.

The waiting time $T^w$ of the task is determined by the virtual machine state chosen by the DRL model. This virtual machine state will be affected by previously executed tasks, especially the end time of the task.

**Table 1 Symbols employed in the scheduling model.**

| Notation | Definition |
|---|---|
| $Q$ | The task queue |
| VMs | The virtual machine group |
| $n$ | The task queue size |
| $m$ | The virtual machine group size |
| $T_i$ | The $i$-th task in the task queue |
| $VM_j$ | The $j$-th virtual machine of a virtual group |
| $T^{res}$ | The response time for the task |
| $T^w$ | The wait time for the task |
| $T^e$ | The execution time for the task |
| $l_i$ | The length of the $i$-th task |
| $T_i^a$ | The arrival time for the $i$-th task |
| $reqC_i$ | The CPU resource request for the $i$-th task |
| $reqM_i$ | The memory resource request for the $i$-th task |
| $reqIO_i$ | The disk I/O resource request for the $i$-th task |
| $T_i^D$ | The deadline for the $i$-th task |
| $T_i^E$ | The end time for the $i$-th task |
| $load_i$ | The load generated when the $i$-th task is executed |
| $V_j$ | The raw speed for the $j$-th virtual machine |
| $load_j$ | The load on the $j$-th virtual machine |
| $C_j$ | The CPU resource of the $j$-th virtual machine |
| $M_j$ | The memory resource of the $j$-th virtual machine |
| $IO_j$ | The disk I/O resource of the $j$-th virtual machine |
| $r_j^t$ | The is the set of tasks running on the $j$-th virtual machine at time $t$ |
| $\theta$ | The load thresholds for virtual machine |

$$T_i^E = T_i^a + T_i^{res} \tag{3}$$

where $T_i^E$ is the end time of the $i$-th task, $T_i^a$ and $T_i^{res}$ are the arrival time and response time of the $i$-th task respectively.

Thereby, the waiting time of the $i$-th task is

$$T_i^w = \begin{cases} 0, & \textit{if first task or condition} \\ \textit{minimize}\{T_k^E | condition\} - T_i^a, & \textit{otherwise} \end{cases} \tag{4}$$

subject to $VM_k = VM_i = j, \ k \in [1, \ i-1]$ (4a)

$T_k^a < T_i^a, \ T_i^a < T_k^E, T_k^{res} \leq T_k^D$ (4b)

where $condition = \textit{fulfill } reqC_i, reqM_i, \ reqIO_i \ and \ (load_j^t + load_i) \leq \ \theta, \ VM_k$ and $VM_i$ are the virtual machines selected by DRL to execute the $k$-th task and the $i$-th task. The above formula $VM_k = VM_i = j$ means that the virtual machines selected by the two tasks are the same, both are the $j$-th virtual machine. And $load_j^t$ is the real-time load of the $j$-th virtual machine at time $t$, and $load_i$ is the load generated by the execution of the $i$-th task.

When a task is run on a virtual machine, the task will produce a certain load on the virtual machine, and the virtual machine load will have a certain impact on the performance of the virtual machine, that is, the processing speed $V$ of the virtual machine. In moment $t$, the $i$-th task commences execution on the selected $j$-th virtual machine. The task will be processed at the real-time processing speed $V_j^t$ corresponding to the real-time load of the virtual machine at moment $t$, thus obtaining the execution time $T_i^e$ of the current task.

$$T_i^e = \frac{l_i}{V_j^t} \tag{5}$$

where $l_i$ is the task length of the $i$-th task, and $V_j^t$ is the real-time processing speed of the $j$-th virtual machine selected at the moment $t$ when the $i$-th task starts running.

## Real-time performance platform model

The real-time performance platform is composed of a group of virtual machines, which can be denoted as VMs = $\{VM_1, VM_2, \cdots, VM_m\}$, where VM represents a virtual machine, and $m$ indicates the total count virtual machines on the current cloud platform. A virtual machine is an independent and isolated computing environment that allows users to feel like they are using a real physical machine. Similar to physical machines, virtual machines include CPU, disk, memory, and processing speed. The processing speed of virtual machines is a crucial factor considered, which is extended to virtual machine performance in this article. The DRL model utilizes current state information to make decisions for each task, assigning them to the most appropriate virtual machine for execution. In this article, the processing speeds of different virtual machines are assumed to be consistent. Still, due to varying loads during task execution, the processing speeds under different loads naturally differ. Meanwhile, load thresholds $\theta$ are set for virtual machines. Once this threshold is exceeded, tasks cannot execute and enter the virtual machine waiting queue until the virtual machine load meets the needs of the task, which is

$$T_i \ running \ status = \begin{cases} Execute, & if \ condition \\ Wait, & otherwise \end{cases} \tag{6}$$

For the literature (*Toumi, Brahmi & Gammoudi, 2022*), the resource contention rate directly affects the server computational performance in the cloud, *i.e.*, when tasks are executed in parallel, the contention rate for the resources increases which leads to the rapid degradation of the server performance. The general linear function is difficult to reflect the complex relationship between load and performance, whereas the power function as a nonlinear function has good nonlinear fitting ability. Consequently, combining the above discussion and considering the mutual interference of parallel execution of tasks, this article expresses the relationship between virtual machine performance and load as follows

$$V_j^t = V_j \times \left(1 - \sqrt{load_j^t}\right) \tag{7}$$

where $V_j^t$ represents the real-time processing speed of the $j$-th virtual machine at time $t$, $V_j$ is the originally defined processing speed of the $j$-th virtual machine, and $load_j^t$ is the real-time load of the $j$-th virtual machine at time $t$.

In this scheduling model, each virtual machine is represented as

$$VM_j = \left\{ V_j, \ load_j, \ C_j, \ M_j, \ IO_j \right\} \tag{8}$$

where $V_j$ is the original definition of the processing speed for the $j$-th virtual machine, and $load_j$ represents the load of the $j$-th virtual machine, with values ranging between [0, 1]. $C_j, M_j, \ IO_j$ respectively represent the CPU, memory and disk I/O resources of the $j$-th virtual machine. For this article, $C_j, M_j, \ IO_j = 1$.

## Real-time load model

According to the current environment status, the DRL model determines the appropriate virtual machines for task allocation and execution. Considering the diversity and dynamics of task demands and the continuity of tasks, the subsequent arrival and execution of tasks are influenced by the impact of previously arrived tasks on the performance of virtual machines. To accurately capture the real server load dynamics, in this article, the virtual machine global load score in *Elsakaan & Amroun (2024)* is adopted as the load caused by task requirements on the virtual machine. Therefore, the load generated by the resource request of CPU, memory, and disk I/O by the $i$-th task is

$$load_i = \alpha * reqC_i + \beta * reqM_i + \gamma * reqIO_i \tag{9}$$
$$subject \ to \quad \alpha + \beta + \gamma = 1 \tag{9a}$$

where $\alpha$, $\beta$, and $\gamma$ are the weights of CPU, memory, and disk I/O resource requests in the generated load respectively.

The main objective of load balancing is evenly distributing tasks among virtual machines within the system. And ensure that the load borne by each virtual machine on the cloud platform is relatively balanced while ensuring the successful execution of tasks, thereby enhancing the overall performance of the system. Consequently, the real-time load of the $j$-th virtual machine at the moment $t$ and the corresponding real-time performance platform average load and load variance are

$$load_j^t = sum\left\{ load_i | T_i \in r_j^t \right\} \tag{10}$$
$$subject \ to \quad T_i^{res} \leq T_i^D, \ T_i^a < t < T_i^E \tag{10a}$$
$$load_{avg}^t = \frac{1}{m} \sum_{j=0}^{m} load_j^t \tag{11}$$
$$load_{var}^t = \frac{1}{m} \sum_{j=0}^{m} \left( load_j^t - load_{avg}^t \right)^2 \tag{12}$$

where $r_j^t$ is the set of tasks running on the $j$-th virtual machine at time $t$, $t \in \left[ T_1^a, \ \cdots, \ T_n^a \right]$.

# THE PROPOSED SRP-DRL

Firstly, we propose the SRP-DRL task scheduling framework and the presentation of the pseudo-code for the DQN-based task scheduling algorithm. Secondly, we elucidate the specific implementation process of the DRL model, including the employed reinforcement learning state enhancement method—real-time performance-aware strategy, the design of particular state-action reward functions, and pseudocode for obtaining task runtime performance parameters.

## SRP-DRL task scheduling framework

The scheduling framework of SRP-DRL is illustrated in Fig. 1. The scheduling framework is composed of four main modules: (a) the task queue, (b) the DRL model, (c) the real-time performance platform model, and (d) the real-time load model. The corresponding workflow of the scheduling model is as follows. First, the user sends requests to the task queue of the scheduling model. Second, the DRL model takes the task information from the task queue as input. Third, the DRL model obtains real-time performance platform server status information. Fourth, DRL makes scheduling decisions built on the status information of the real-time performance platform and assigns a virtual machine to the task. Fifth, the task runs on the selected virtual machine and evaluates the task running performance parameters under the deadline (DDL) conditions. Simultaneously, the real-time load state of the virtual machine group before the real-time performance platform executes the task is fed into the load model. Finally, the DRL model obtains the reward value returned depending on a comprehensive evaluation of the task running performance parameters under the DDL conditions and the real-time load state, which guides the model's training.

We choose the DRL model DQN to complete task scheduling, reduce task response time, and improve server load balancing capability while considering the server's real-time performance and delay constraints. DQN is an effective algorithm in DRL. In utilizing deep neural networks, the method aims to approximate the optimal action-value function $Q^*$, aiming to obtain the optimal policy for maximizing cumulative rewards.

$$Q(s_t, a_t) \leftarrow Q(s_t, a_t) + \alpha * \left( r_t + \gamma * \max_{a \in A} Q(s_{t+1}, a) - Q(s_t, a_t) \right) \tag{13}$$

where $Q(s_t, a_t)$ represents the $Q$-value of executing action $a_t$ under state $s_t$, $\alpha$ is the learning rate, $r_t$ is the immediate reward obtained by the agent executing action $a_t$, $\gamma$ is the discount factor, $s_{t+1}$ is the newly observed environmental state after executing action $a_t$, $A$ denotes the action space of the DRL model, and $\max_{a \in A} Q(s_{t+1}, a)$ corresponds to the action value of the action that maximizes the $Q$-value in state $s_{t+1}$. Algorithm 1 describes the pseudocode for task scheduling based on DQN.

## DRL model

### Real-time performance-aware strategy

Server load affects the server's performance, and the server's real-time performance directly impacts the efficiency and performance of cloud task execution. In this article, we

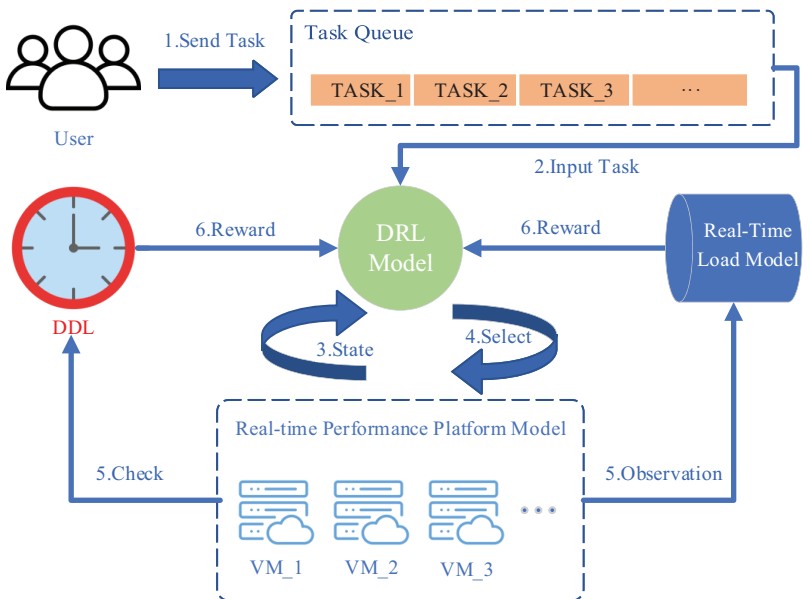

**Figure 1 Scheduling framework.** Image credits: server free icon (created by shin_icons; Flaticon); clock free icon (created by IYIKON; Flaticon); Group free icon (created by Prosymbols; Flaticon).

use the weighted resource requests of tasks as the server load and the server's processing speed as the performance representation of the server. In the real-time performance platform modeling part of "System Model", the relationship between server load and performance is defined as a power function model with a nonlinear relationship. Taking into account the mutual interference of parallel execution tasks, the execution speed of the server decreases rapidly as the load increases.

To conduct a more detailed and comprehensive modeling of the state of the currently proposed task scheduling environment that considers both server load and the impact of server load on server performance, we normalize the real-time speed of the virtual machine at time $t$ and add the normalized results to the state space composed of virtual machine load information at time $t$. This augmentation aims to enhance the perceptual capabilities of the DRL model.

$$nor_{V^t} = \frac{V^t - minV^t}{max(\max V^t - minV^t, 1)} \tag{14}$$

where $V^t$ is the real-time processing speed of the virtual machine group at time $t$, which is a vector with a length of $m$, and $nor_{V^t}$ is the normalized real-time processing speed of the virtual machine group at time t, with values ranging from 0 to 1.

### Detailed design of SRP-DRL

In implementing cloud task scheduling using DRL algorithms, the design of the state space, action space, and reward function is crucial, as these factors will directly impact the performance of the algorithmic model. The SRP-DRL method model is elaborated upon in detail, and its model diagram is depicted in Fig. 2. Furthermore, the algorithm pseudo-code

| Algorithm 1 Task scheduling based on DQN. |
|---|

**Input:** Q // Q is task queue

**Output:** Average load variance, task success rate, and average response time

1    Initialize the environment and set parameters;

2    **for** epoch **do**

3        Get state $s_t$ include real-time performance and load;

4        **for** each $T_i$ in Q **do**

5            Select an action $a_t$ based $\varepsilon$-greedy strategy,

6            otherwise $a_t = \arg\max_{a \varepsilon A} Q(s_t, a; \theta)$;

7            Execute action $a_t$ and **PerformanceParametersGeneration ($T_i$, $a_t$)**;

8            Obtain immanent reward $r_t$ and new environment state $s_{t+1}$;

9            Store experience $(s_t, a_t, r_t, s_{t+1})$ to experience replay buffer;

10           **if** number of tasks executed > learning threshold:

11               Random sampling batch experience using Eq. (13);

12               Update target network parameters after every $\delta$ rounds of learning;

13       **End do**;

14   **End do**;

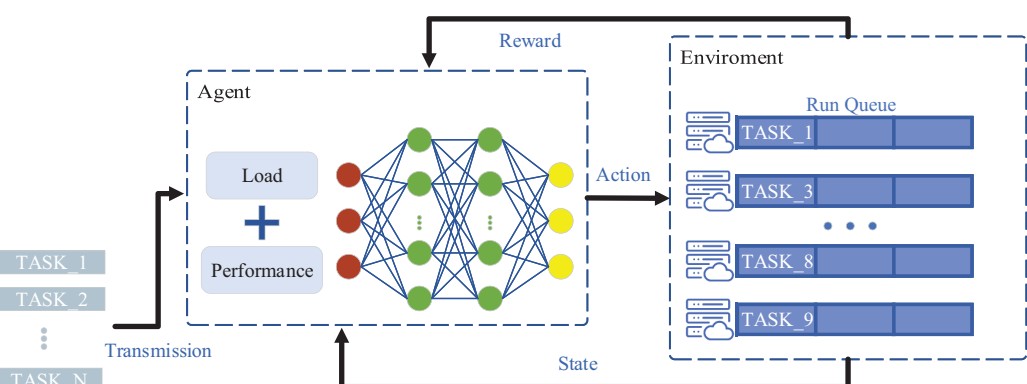

**Figure 2 SRP-DRL model.** Image credits: server free icon (created by shin_icons; Flaticon)

is presented for SRP-DRL to obtain task execution performance parameters based on real-time load and performance.

**State space.** The state space involved in the reinforcement learning model is crucial for the performance and adaptability of the algorithm. The proposed SRP-DRL algorithm considers the effect of server load on performance and introduces a real-time performance-aware strategy to enhance the model's perception of environmental states. In the SRP-DRL algorithm at time $t$, the model's state space $S$ is defined as

$$s_t = \left\{ V_1^t, V_2^t, \cdots, V_m^t, load_1^t, load_2^t, \cdots, load_m^t \right\}.$$

---

**Algorithm 2** PerformanceParametersGeneration.

**Input:** $T_i$, $a_t$, historical task performance parameters, and historical task run queue

**Output:** Task running performance parameters

1    Compute the $load_i$ occupied by $T_i$ execution;

2    // $\theta$ is load threshold

3    Gain real-time load of virtual machine $load_j^t$ based on $a_t$;

4    **if** first task OR (fulfill $reqC_i$, $reqM_i$, $reqIO_i$ AND ($load_j^t + load_i$) $\leq \theta$):

5        Obtain task running performance parameters;

6    **else**:

7        Get the task running queue of the virtual machine;

8        Arrange the task running queue in ascending order according to task end time;

9        Calculate task execution time $T_i^e$;

10      // Traverse the task running queue

11      **While** unfulfilled $reqC_i$, $reqM_i$, $reqIO_i$ OR ($load_j^t + load_i$) $> \theta$:

12          Update **$load_j t$, remaining resources and $T_i^e$**;

13      Calculate task running performance parameters;

---

**Action Space.** The action space of a model is the aggregation of virtual machines from which tasks can be selected, specifically expressed as $A = \{VM_1, VM_2, \cdots, VM_m\}$. The action decision output of the model is a one-hot encoded value in the action space, that is, one for the virtual machine selected by the model for the task and 0 for the rest. For example, when the $m$-th virtual machine is selected, then $a = \{0, 0, \cdots, 1\}$.

**Reward Function.** To guide the reinforcement learning agent toward optimizing load balancing, the reward function is defined as

$$R = \begin{cases} -1, & \text{Violation of DDL} \\ \dfrac{0.1}{T^{res}}, & \text{Not violating DDL and } load_j^t > load_{avg}^t \\ \dfrac{\left| load_j^t - load_{avg}^t \right|}{T^{res}}, & \text{Not violating DDL and } load_j^t \leq load_{avg}^t \end{cases} \tag{15}$$

where $T^{res}$ represents the response time of task execution, $load_j^t$ is the real-time load of the $j$-th virtual machine selected by the task at moment $t$ before executing the current task, and $load_{avg}^t$ is the average load of the real-time performance platform at the moment $t$ when the current task is not executed.

In SRP-DRL, the real-time load and performance of the server dynamically change every time, which means that the operating performance parameters such as response time, load, *etc.*, for the same task under different server loads will also be different. Hence, it is necessary to obtain the running performance parameters of the task depending on the real-time load and performance state of the server where the task is selected for execution

**Table 2 Experimental data generation process.**

| $T_i$ Attributes | Obey distribution |
|---|---|
| $l_i$ | $N(200,\ \sigma^2)$ |
| $T_i^a$ | $P(\lambda)$ |
| $reqC_i$ | $U[0,\ \theta)$ |
| $reqM_i$ | |
| $reqIO_i$ | |

to instruct the model training. Algorithm 2 describes the pseudocode of the SRP-DRL method to obtain task running performance parameters.

# EXPERIMENT

This section begins with an explanation of the experimental parameter settings and data generation process and then describes the performance evaluation metrics and baseline algorithms for experimental validation. Finally, the effectiveness of the proposed method is verified by comparing it with four other task scheduling methods (Random, Time-Slice Rotation Round-Robin, Earliest Idle Time First-EITF, and BEST-FIT) in four dimensions: various number of tasks, varying task arrival rates, different numbers of virtual machines, and diverse task length standard deviations.

## Parameter settings and data generation

In our experiments, the weighted resource requests of tasks is considered as the load generated when tasks run on virtual machines. In the literature (*Cheng et al., 2022*), the authors chose to use a two-layer neural network model to approximate the optimal action-value function based on simplicity and computational efficiency, and the results show that this choice can improve scheduling efficiency. To make full use of the advantages of this model, it is introduced into our research, and at the same time, the overall load balancing performance can be improved after making necessary hyperparameter adjustments. The experience replay buffer size is 800, the batch size for mini-batch stochastic gradient descent is 60, the learning rate is 0.001, the experience replay learning threshold is 500, and the target network parameter update frequency is 50. For the cloud platform, the virtual machine load threshold $\theta$ is 0.95, the initial speed of virtual machines is 500. Regarding the task execution process, the most crucial thing is the deadline of the task, which in the experiments was 0.5, dictated by the average length of tasks and the original processing speed of virtual machines.

The relevant attributes of the randomly generated tasks in the experiment are all subject to specific probability distributions, as shown in Table 2. The task length $l_i$ obeys a normal distribution with an expectation of 200 and a variance of $\sigma^2$; the task arrival time $T_i^a$ obeys a Poisson distribution with a parameter of $\lambda$, $\lambda$ is expressed as the task arrival rate in the text; the resource requests of the task $reqC_i$, $reqM_i$, $reqIO_i$ all

obey the uniform distribution with a lower limit of 0 and an upper limit of the load threshold $\theta$.

## Evaluation indicators

In the experiment, we evaluate the performance of the SRP-DRL scheduling method from task average response time, task success rate, and server average load variance.

**Task average response time.** Response time is an essential indicator for measuring load balancing. It can evaluate the server running status and performance of the real-time performance platform. It is defined as follows.

$$T_{avg}^{res} = \sum_{i=0}^{n} T_i^{res} \tag{16}$$

**Task success rate.** Whether the task responds successfully depends on whether the remaining resources and load status of the current server are sufficient to meet the task's resource requirements and whether the task's response time can meet the task's DDL constraints, that is, $T^{res} \leq T^D$. The task success rate can be expressed as

$$TSR = \frac{task_{suc}}{n} \tag{17}$$

where, $task_{suc}$ is the number of successfully executed tasks, and $n$ is the total number of task requests in the task queue.

**Server average load variance.** Load variance can detect the fluctuation degree of server load and intuitively reflect the load balancing effect of the real-time performance platform. Under the condition of ensuring a high task success rate, the smaller the load variance, the more balanced the server load. On the contrary, it means the load is more unbalanced, which is defined as

$$avg\_load_{var} = \frac{1}{n} \sum_{i=0}^{n} load_{var}^t \tag{18}$$

## Baseline algorithm

In this article, the load-balancing performance of SRP-DRL is evaluated by comparing it to four baseline algorithms.

**Random.** The task randomly selects a virtual machine for execution. The algorithm is simple and fast, but task execution performance cannot be guaranteed.

**Round-Robin.** Tasks are assigned to virtual machines in turn in order. The algorithm allocates resources fairly but may result in poor resource utilization.

**EITF.** The task selects the earliest idle virtual machine to execute the task. The algorithm reduces waiting times but may result in less efficient use of idle resources.

**BEST-FIT.** The task dynamically selects the most appropriate virtual machine for execution based on task requirements and virtual machine status. The algorithm can effectively utilize resources, but the implementation is more complex and may increase system overhead.

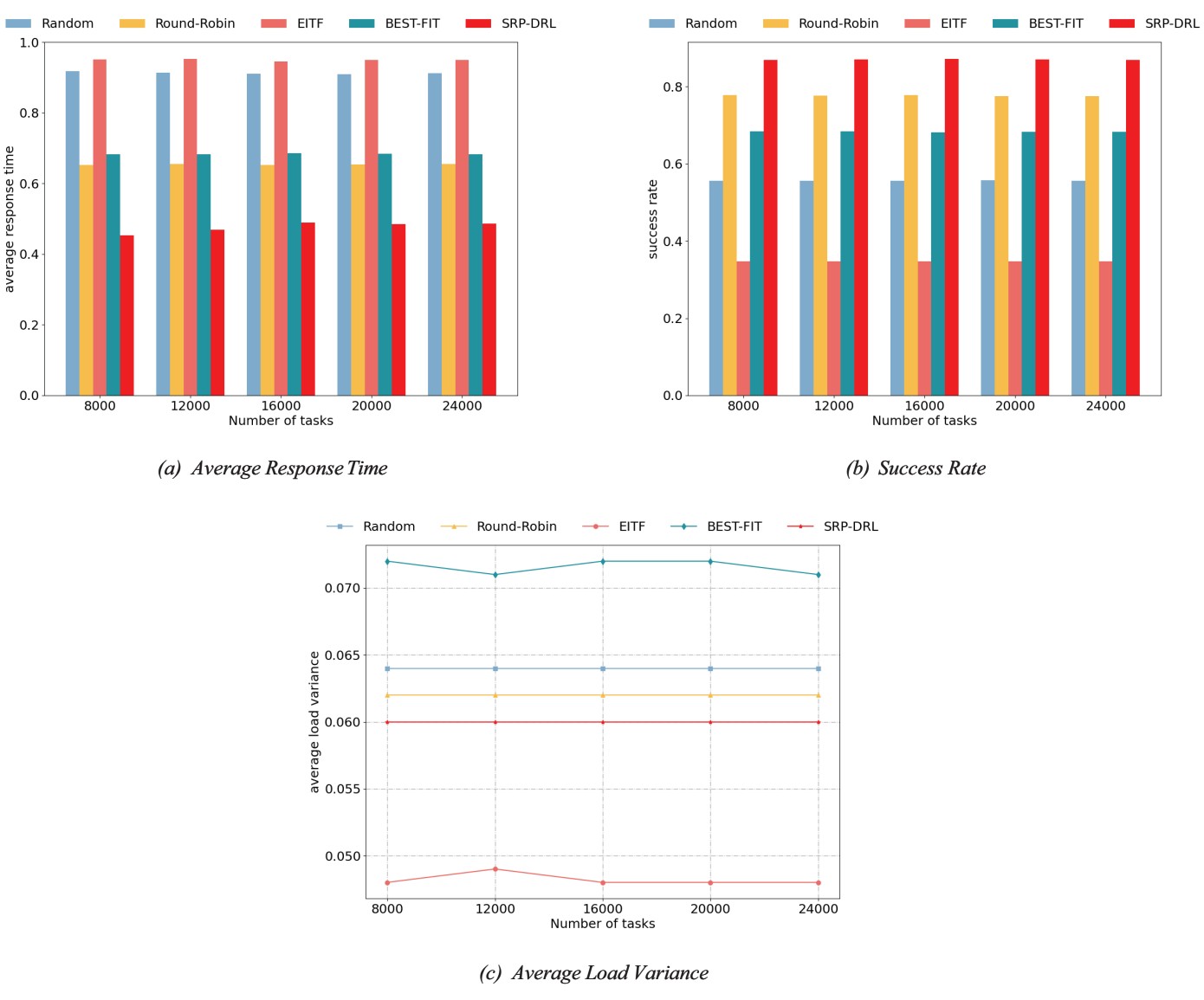

*(a) Average Response Time*

*(b) Success Rate*

*(c) Average Load Variance*

**Figure 3 Comparison of load balancing performance of scheduling methods under different number of tasks.**

## Comparison results and analysis

**Different number of tasks.** At this experiment, we compare the load-balancing performance of five scheduling methods with varying numbers of tasks. The number of tasks is increased from 8,000 to 24,000 with an interval of 4,000. The task arrival rate is 20, the number of virtual machines is 10, and the standard deviation $\sigma$ of the task length is 20. The results of the experiment are shown in Fig. 3.

Concretely, Fig. 3A illustrates the task response time, Fig. 3B presents the task success rate, and Fig. 3C shows the average load variance. In Figs. 3A–3C, as the number of tasks increases, the average task response time, task success rate, and average virtual machine

load variance do not change significantly. Experiments show that the number of tasks has little impact on the load-balancing performance of the current five task scheduling variances. As the experiment results are shown in Fig. 3, SRP-DRL is significantly better than other task scheduling methods in average response time and success rate. However, the average load variance of the EITF algorithm is lower than that of SRP-DRL. This is because the success rate of the EITF algorithm is very low, resulting in slight load fluctuation on the virtual machine during the task scheduling process, ultimately leading to a lower average load variance for the EITF algorithm. Hence, the comprehensive load balancing performance of SRP-DRL is better than that of other task scheduling methods.

**Different task arrival rates.** For this experiment, the load-balancing performance of five scheduling methods was compared under diverse task arrival rates. The task arrival rate ranged from 20 to 40 with intervals of 5, the number of virtual machines was 10, the number of tasks was 8,000, and the standard deviation of task length $\sigma$ was 20. The findings of the present experiment are shown in Fig. 4.

The quantity of tasks arriving within a specific time frame is determined by the task arrival rate. The higher the task arrival rate, the more tasks come in a unit of time. In particular, Fig. 4A indicates task response time, Fig. 4B shows task success rate, and Fig. 4C represents average load variance. As shown in Figs. 4A–4C, with the increase in task arrival rate, the average task response time slowly increases, the task success rate is gradually decreasing, and the average load variance of the virtual machine is gradually decreasing except for Random and EITF. This is because with more tasks arriving in a unit of time and the limited processing capacity of virtual machines, the waiting time for tasks increases, affecting task response time and ultimately leading to task failure beyond the deadline. Increasing the number of tasks arriving in a unit of time allows for a more effective distribution of tasks among virtual machines, avoiding overloading or underloading certain virtual machines, thereby affecting the average load variance of the entire virtual machine group. Specifically, the average load variance of the SRP-DRL algorithm is lower than that of the EITF algorithm at a task arrival rate of 40. As shown in the experimental results in Fig. 4, the comprehensive load balancing performance of SRP-DRL compares favorably with other task scheduling methods in the three indicators of average response time, success rate, and average load variance.

**Different number of virtual machines.** In this experiment, the study compared the load-balancing performance of five scheduling methods with other numbers of virtual machines. The number of virtual machines increased from 10 to 18 with an interval of 2, the task arrival rate was 40, the number of tasks was 8,000, and the task length standard deviation $\sigma$ was 20. The performance of the experimental results is shown in Fig. 5.

For details, Fig. 5A presents task response time, Fig. 5B illustrates task success rate, and Fig. 5C indicates the average load variance. From Figs. 5A–5C, with the increase in the number of virtual machines, the average task response time decays gradually, the task success rate increases. In contrast, the average load variance of virtual machines increases slowly, except for the fluctuation of Random and EITF. For this reason, as virtual machines increase, the system's ability to process tasks increases, reducing task wait times and affecting average task response time and success rate. However, maintaining load balance

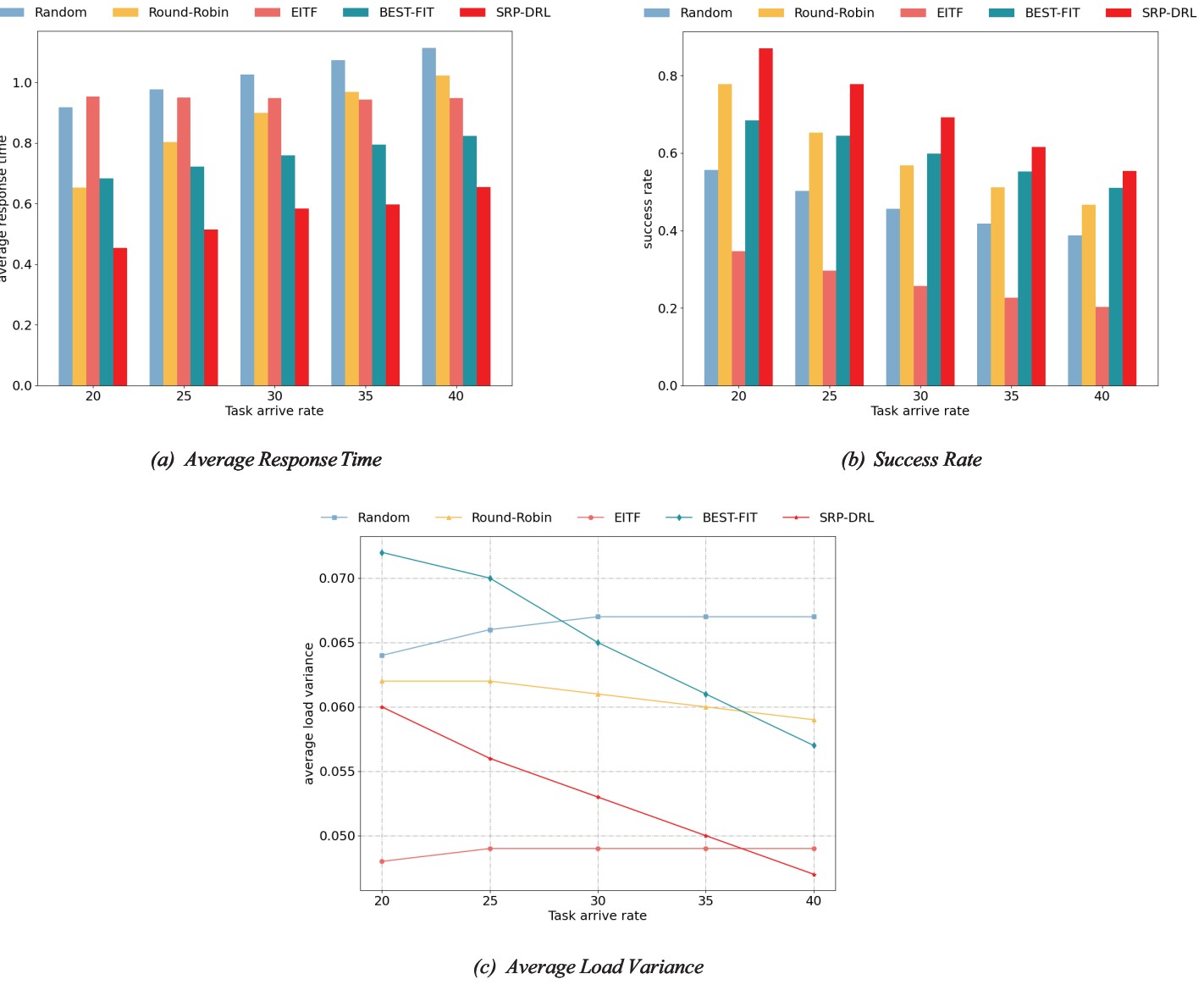

*(a) Average Response Time*

*(b) Success Rate*

*(c) Average Load Variance*

**Figure 4 Comparison of load balancing performance of scheduling methods under different task arrival rates.**

between virtual machines becomes increasingly challenging, leading to fluctuations in the average load variance of the virtual machine group. It is shown in Fig. 5 of the experimental results that SRP-DRL also outperforms other task scheduling methods in terms of combined load balancing performance regarding average response time, success rate, and average load variance.

**Different task length deviations.** Within this experiment, a comparative study on the load-balancing performance of five scheduling methods was conducted for various task length standard deviations. The task length standard deviation $\sigma$ ranged from 10 to 50 with

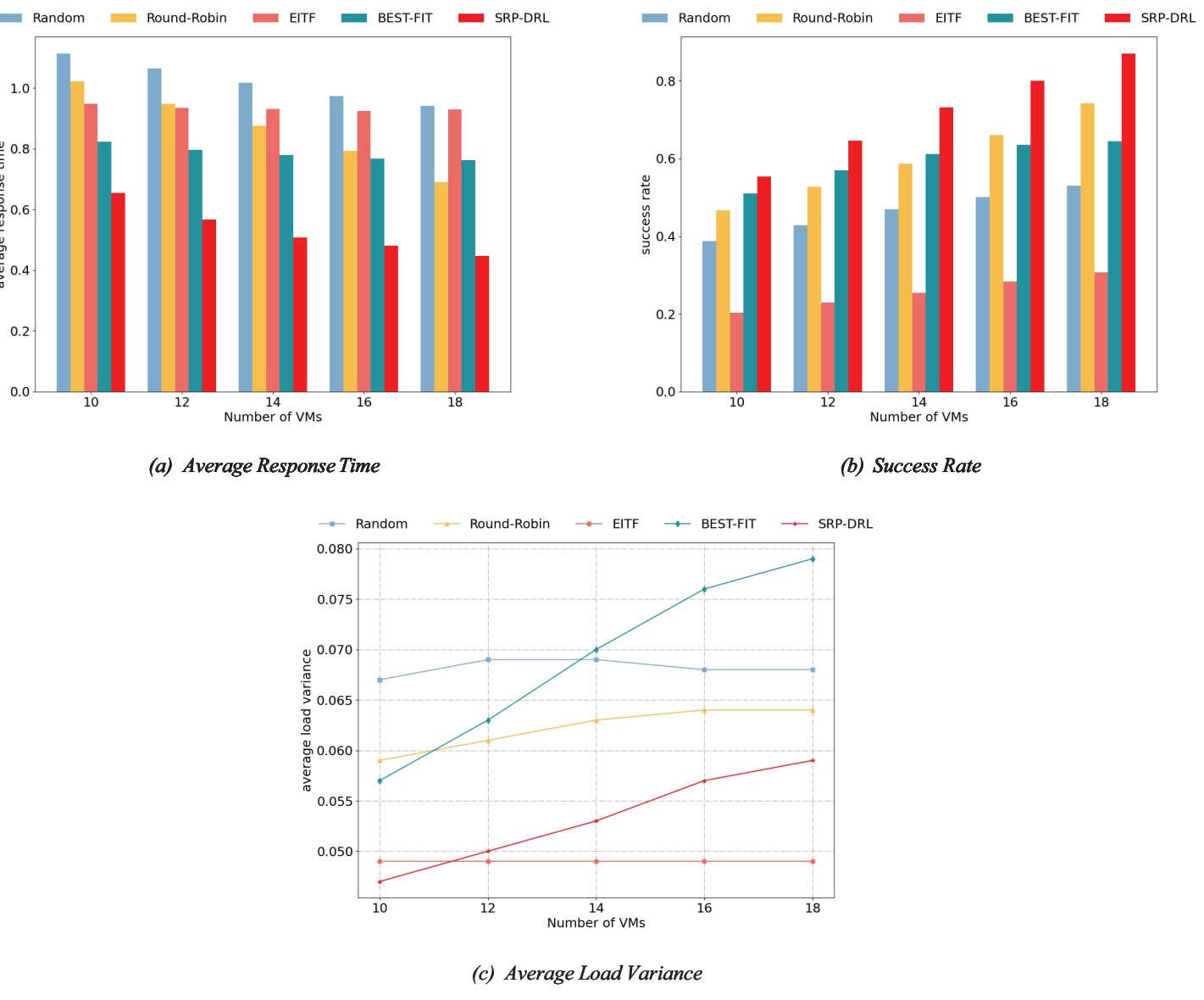

*(a) Average Response Time*

*(b) Success Rate*

*(c) Average Load Variance*

**Figure 5 Comparison of load balancing performance of task scheduling methods under different numbers of virtual machines.**

intervals of 10, the task arrival rate was 30, the number of tasks was 8,000, and the number of virtual machines was 10. The outcome of the experiment is shown in Fig. 6.

Specifically, Fig. 6A indicates task response time, Fig. 6B shows task success rate, and Fig. 6C illustrates the variance in average load. In Figs. 6A–6C, with the increase in task length standard deviation, the average load variance of virtual machines gradually increases except for Random. The cause of this is task lengths become more diverse with an increasing standard deviation of task lengths. The load generated by task execution fluctuates wildly in resource occupancy time (that is, task execution time), making it

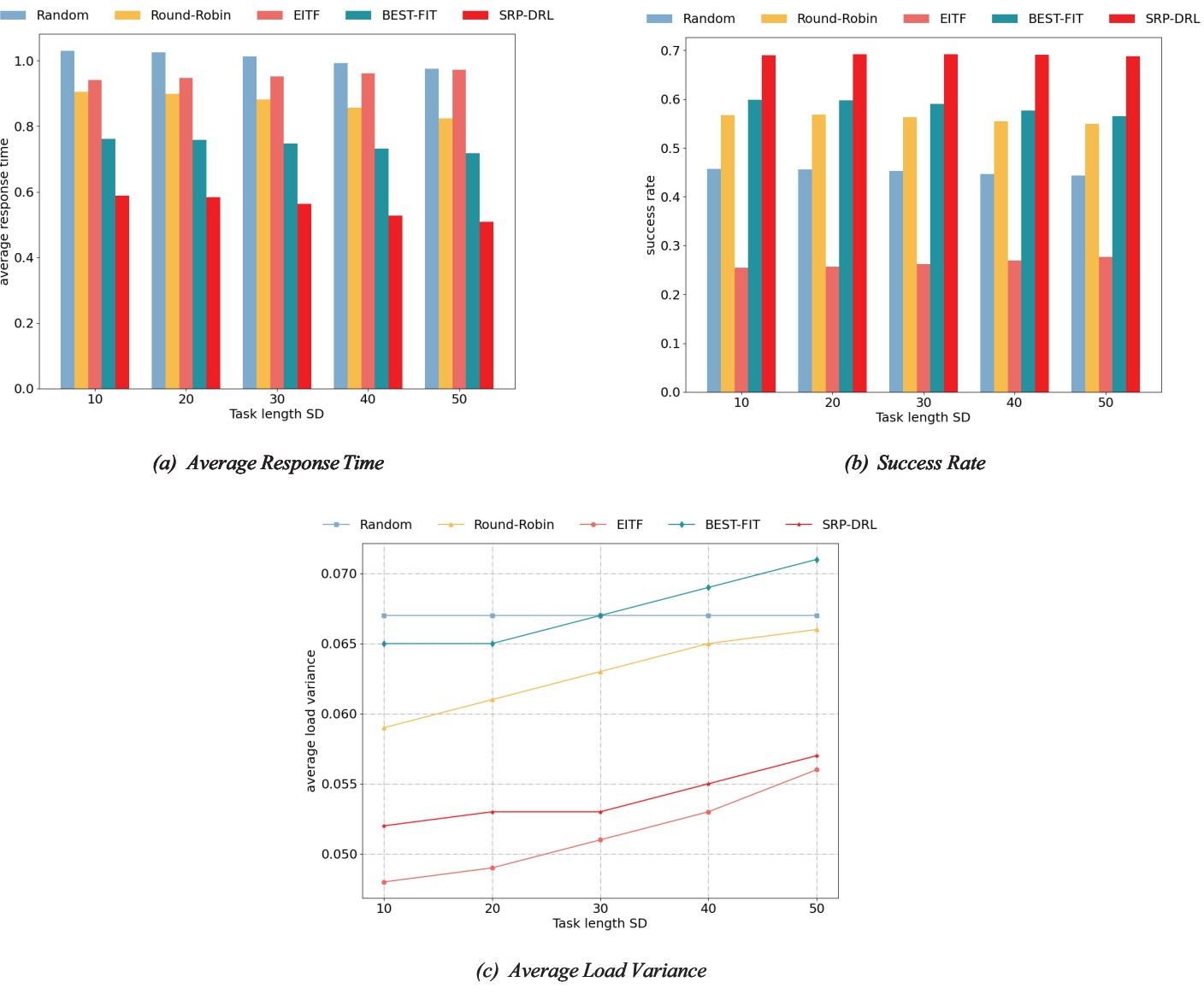

*(a) Average Response Time*

*(b) Success Rate*

*(c) Average Load Variance*

**Figure 6** Comparison of load balancing performance of task scheduling methods under different task length standard deviations.

increasingly difficult to maintain load balancing among virtual machines. This is shown in Fig. 6 of the experimental results, which shows that the integrated load balancing performance of SRP-DRL was similarly superior to other task scheduling methods in all three metrics of average response time, success rate, and average load variance. Indeed, the performance of EITF on the average load variance metric is consistent with the results in Fig. 3C.

## CONCLUSION

This article employs a DRL approach to optimize the load balancing issue in cloud task scheduling. Considering the effect of server load on performance, we propose a DRL task

scheduling method based on server real-time performance SRP-DRL. Under latency constraints, this method reduces task response time and enhances server load balancing capability. Experimental results indicate that compared to four baseline scheduling methods, the comprehensive load balancing performance of SRP-DRL outperforms them regarding average response time, task success rate, and average load variance.. For example, when the number of virtual machines is 10, the number of tasks is 8,000, the standard deviation of task length $\sigma$ is 20, and the task arrival rate is 40, compared with Random, Round-Robin, EITF and BEST-FIT, SRP-DRL achieves the average load variance is reduced by 29.9%, 20.3%, 4.1%, and 17.5%.

However, the relationship between server load and performance is highly complex. Metrics measuring server loads, such as CPU utilization, memory usage, and disk I/O, influence server performance. Accurately measuring and mapping server load to performance remains a challenging problem. Therefore, in future work, we will emphasize the precise measurement and mapping relationship between server load and performance, improving the quality of the reinforcement learning environment for better optimization of cloud task scheduling issues. In addition, although this article uses a two-layer neural network model with fixed parameters, which is beneficial to improving load balancing performance, it may limit the flexibility and adaptability of the DRL algorithm in some particular scenarios. Therefore, in the future, we will also focus on different network architectures or hyperparameter settings to improve the DRL algorithm's performance further.

### Funding
This work was supported by the National Natural Science Foundation of China (No. 52275480), the Science and Technology Project of Guizhou Provincial Department (No. QKHZYD[2023]002). There was no additional external funding received for this study. The funders had no role in study design, data collection and analysis, decision to publish, or preparation of the manuscript.

### Grant Disclosures
The following grant information was disclosed by the authors:
National Natural Science Foundation of China: 52275480.
Science and Technology Project of Guizhou Provincial Department: QKHZYD[2023]002.

### Competing Interests
The authors declare that they have no competing interests.

### Author Contributions

- Jinming Wang conceived and designed the experiments, performed the experiments, analyzed the data, performed the computation work, prepared figures and/or tables, authored or reviewed drafts of the article, and approved the final draft.

- Shaobo Li conceived and designed the experiments, authored or reviewed drafts of the article, and approved the final draft.
- Xingxing Zhang analyzed the data, prepared figures and/or tables, authored or reviewed drafts of the article, and approved the final draft.
- Fengbin Wu analyzed the data, prepared figures and/or tables, authored or reviewed drafts of the article, and approved the final draft.
- Cankun Xie performed the computation work, prepared figures and/or tables, and approved the final draft.

## Data Availability

The data utilized in this article is generated randomly during code execution, without reliance on or analysis of third-party data. The original experimental results and code are available in the Supplemental File.

## Supplemental Information

Supplemental information for this article can be found online at http://dx.doi.org/10.7717/peerj-cs.2120#supplemental-information.

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
