# Peer review of "Deep reinforcement learning task scheduling method based on server real-time performance"

_PeerJ Computer Science, doi:10.7717/peerj-cs.2120_

## Round 0.1 · original submission · Major Revisions

Based on the reviewers’ comments, you may resubmit the revised manuscript for further consideration. Please consider the reviewers’ comments carefully and submit a list of responses to the comments along with the revised manuscript.

**Language Note:** PeerJ staff have identified that the English language needs to be improved. When you prepare your next revision, please either (i) have a colleague who is proficient in English and familiar with the subject matter review your manuscript, or (ii) contact a professional editing service to review your manuscript. PeerJ can provide language editing services - you can contact us at copyediting@peerj.com for pricing (be sure to provide your manuscript number and title). – PeerJ Staff

Reviewer 1 ·

Basic reporting

I had some minor feedback with respect to the structure of this article. It would appear that the figures, graphs, and tables are provided at the end of the article after the list of references. I would like to request the authors to please move them to be as close to the referring text as possible.

Experimental design

no comment

Validity of the findings

no comment

Additional comments

This article presents a real-time performance-aware strategy for task scheduling on cloud servers. Based on real-time server performance, a distributed reinforcement learning technique for scheduling is proposed that optimizes task response time and improves load balancing in the system.

Overall, I liked the article. Specifically:
- The article is well organized.
- The authors have done a good job in providing the necessary background to understand the article.
- The system model section provides the reader of a good theoretical understanding of the problem as well as the assumptions and definitions involved
- The experimental evaluation is well done

My only feedback is around the location of the figures, graphs, and tables. I would like to request the authors to please move them to be as close to the referring text as possible.

Reviewer 2 ·

Basic reporting

The paper addresses the significant challenge of load balancing in cloud task scheduling, a critical issue in cloud computing environments. It provides a clear and comprehensive background on the problem domain, highlighting the importance of optimizing server performance and reducing task response times in cloud-based systems.
The literature review is adequately structured and covers relevant research on load balancing and task scheduling in cloud environments. However, it could benefit from further exploration of recent advancements in deep reinforcement learning (DRL) approaches for addressing similar challenges.
The paper appropriately acknowledges the importance of data availability, indicating that the data used in the experiments is generated randomly during code execution. However, more detailed information on the experimental data generation process and potential biases would enhance the transparency of the research.
Overall, the paper effectively communicates the motivation and context of the study, providing readers with a solid foundation for understanding the proposed SRP-DRL task scheduling framework and its implications for addressing load-balancing issues in cloud computing.

Experimental design

The experimental design considers only the execution time of tasks as the server load. This oversimplified representation may not accurately capture the real-world server load dynamics, which typically involve multiple factors such as CPU utilization, memory usage, and disk I/O.
The use of a two-layer neural network model with fixed parameters may limit the flexibility and adaptability of the DRL algorithm. Different network architectures or hyperparameter settings could potentially lead to improved performance.
The experimental settings maintain fixed values for certain parameters such as the number of tasks, task arrival rates, and task length standard deviations across experiments. Introducing more variability in these parameters could provide a more comprehensive understanding of the method's performance under diverse conditions.
The study assumes a linear relationship between server load and performance, which may not accurately reflect the complex and nonlinear nature of server behavior. The simplified model may overlook important nuances in server performance dynamics.
While the study compares the proposed SRP-DRL method against three baseline scheduling methods, the comparison is limited to a single benchmark scenario. Including additional benchmarks or real-world datasets could provide a more robust evaluation of the method's effectiveness.
The experimental design primarily focuses on evaluating the proposed method's performance under specific settings and conditions. The generalizability of the findings to other cloud computing environments or task scheduling scenarios may be limited.

Validity of the findings

The findings of the paper suggest that the proposed SRP-DRL (Server Real-time Performance based Deep Reinforcement Learning) method effectively addresses the load balancing challenges in cloud task scheduling. Through experimental validation, SRP-DRL demonstrates superior performance compared to three baseline scheduling methods: Random, Round-Robin, and Earliest Idle Time First (EITF). Specifically, SRP-DRL outperforms these methods in terms of task success rate, average response time, and average load variance.

For example, when evaluated against the baselines with specific parameters (e.g., 10 virtual machines, 8000 tasks, task length standard deviation of 20, and task arrival rate of 40), SRP-DRL achieves significant reductions in average load variance. These reductions indicate improved load-balancing capabilities, leading to better resource utilization and overall system performance.

Overall, the findings suggest that SRP-DRL holds promise as an effective approach for optimizing cloud task scheduling under latency constraints. However, the paper also highlights the complexity of the relationship between server load and performance, indicating a need for further research to refine the method and address these challenges comprehensively.

Additional comments

The authors need to add some more recent papers to the literature.

---

## Round 0.2 · accepted · Accept

Congratulations, the reviewers are satisfied with the revisions and recommend accept decision.

Reviewer 1 ·

Basic reporting

no comment

Experimental design

no comment

Validity of the findings

no comment

Additional comments

The authors have provided a thorough response to the reviews. I would like to thank them for taking the time and effort to do so.

Reviewer 2 ·

Basic reporting

This article presents a real-time performance-aware strategy for task scheduling on cloud servers. Based on real-time server performance, a proposed distributed reinforcement learning technique for scheduling optimizes task response time and improves load balancing in the system.
Overall, I liked the article. Specifically:
- The article is well organized.
- The authors have done a good job of providing the necessary background to understand the article.
- The system model section provides the reader with a good theoretical understanding of the problem as well as the assumptions and definitions involved
- The experimental evaluation is well done

Experimental design

The literature review is adequately structured and covers relevant research on load balancing and task scheduling in cloud environments. However, it could benefit from further exploration of recent advancements in deep reinforcement learning (DRL) approaches for addressing similar challenges.

Validity of the findings

Satisfactory